Comparison design and evaluation power in cohort and self-controlled case series designs for post-authorization vaccine safety studies

Sato Shuntaro shuntaro.sato.stat@gmail.com 1
Kawazoe Yurika 1
Katsuta Tomohiro 2
Fukuda Haruhisa 3 4
1 Clinical Research Center, Nagasaki University , Nagasaki , Japan
2 Department of Pediatrics, St. Marianna University School of Medicine , Kawasaki , Japan
3 Department of Health Care Administration and Management, Graduate School of Medical Sciences, Kyushu University , Fukuoka , Japan
4 Center for Cohort Studies/Graduate School of Medical Sciences, Kyushu University , Fukuoka , Japan
Suner Aslı
Electronic publication date: 2024 Jan 23
Publication date: 2024
Volume: 12
Electronic Location ID: e16780
Received 2023 Aug 15; Accepted 2023 Dec 18
Copyright: ©2024 Sato et al.
Copyright year: 2024
Copyright holder: Sato et al.
License: This is an open access article distributed under the terms of the Creative Commons Attribution License, which permits unrestricted use, distribution, reproduction and adaptation in any medium and for any purpose provided that it is properly attributed. For attribution, the original author(s), title, publication source (PeerJ) and either DOI or URL of the article must be cited.
License URL: https://creativecommons.org/licenses/by/4.0/

Keywords: Post-authorization vaccine safety studies, Power analysis, Self-controlled case series design

Funding: Japan Agency for Medical Research and Development (AMED) JP21nf0101635 This research was supported by the Japan Agency for Medical Research and Development (AMED) under grant number JP21nf0101635. The funders had no role in study design, data collection and analysis, decision to publish, or preparation of the manuscript.

==============================
Background

Post-authorization safety studies (PASSs) of vaccines are important. PASSs enable the evaluation of association between vaccination and adverse events following immunization through common study designs. Clinical trials during vaccine development typically include a few thousand to 10,000 participants while a PASS might aim to detect a few adverse events per 100,000 vaccine recipients. While all available data may be utilized, prior consideration of power analyses are nonetheless crucial for interpretation in cases where statistically significant differences are not found.

Methods

This research primarily examined cohort study design and self-controlled case series (SCCS) design, estimating the power of a PASS under plausible conditions.

Results

Both the cohort study and SCCS designs necessitated large sample sizes or high event counts to guarantee adequate power. The SCCS design is particularly suited to evaluating rare adverse events. However, extremely rare events may not yield sufficient occurrences, thereby resulting in low power. Although the SCCS design can more efficiently control for time-invariant confounding in principle, it solely estimates relative measures. A cohort study design might be preferred if confounding can be adequately managed as it also estimates absolute measures. It may be an easy decision to use all the data at hand for either design. We found it necessary to estimate the sample size and number of events to be used in the study based on a priori information and anticipated results.

Introduction

The evaluation of vaccine safety is a critical aspect of public health. In the context of post-authorization vaccine safety studies (PASS), where extremely low outcome events are a concern, there is insufficient literature that focuses on sample size or power for cohort and self-controlled case series (SCCS) designs. The guidelines from the European Medicines Agency emphasize the necessity for an a priori sample size calculation without restricting the study design (European Medicines Agency, 2012). Gault et al. (2017) reviewed reports on SCCS designs and observed that few studies include a priori sample size calculations, highlighting the significance of power calculations, even when conducted a posterior. A typical PASS for vaccines is focused on detecting rare adverse events following immunization (AEFIs), which often go unobserved in clinical trials. As Phase III trials in vaccine development typically engage several thousand to ten thousand participants, PASS specifically targets AEFIs with incidence frequencies fewer than several per thousand or even several per ten thousand. While leveraging existing data permits the utilization of all available information, it is often prudent to calculate the sample size a priori, based on the incidence of AEFIs and the expected effect size. Without an a priori sample size determination, it becomes challenging to discern whether an effect is close to null from an inadequate sample size or represents the actual absence of the effect. This ambiguity consequently clouds the interpretation of any results. Therefore, a priori contemplation of sample size or power before undertaking a PASS is crucial.

The present study focuses on the cohort study design as the traditional design and the SCCS design as the case-only design. Both study designs have advantages and are used in PASS. There are major differences between cohort and SCCS studies in the association measures that can be estimated and how they control for confounding. First, as association measures, SCCS studies can only estimate relative measures, while cohort studies can calculate both absolute and relative measures. For public health reasons, it is desirable to be able to calculate both types of measures, as in cohort studies. Second, in terms of controlling for confounding, cohort studies can only control for measurable confounders. SCCS studies, on the other hand, can automatically control for factors that vary little over time (e.g., gender, genetic information).

Our objective was to evaluate the statistical power of these designs within the framework of PASS across various scenarios, including different prevalences, matching ratios, effect sizes, sample sizes, and event numbers.

Materials & Methods

In this study, we concentrated on the cohort study design and the SCCS design as the most applicable approaches for a PASS. Figure 1 illustrates the concept for both the cohort study design and the SCCS design; the SCCS design population constitutes a subset of the cohort study design population. In PASS, the cohort study design is frequently employed due to its traditional framework, which necessitates information on both the outcomes and the exposure. Although case-control studies also rely on this information, we did not addressed this design in this study due to their infrequent use (Sultana et al., 2022).

Figure 1 Design summary of the cohort study and the self-controlled case-series.

This figure showed the groups or time periods compared to the analysis population for the cohort study design and the self-controlled case series design. The self-controlled case series design’s analysis population is part of the cohort study’s analysis population.

Cohort study design

Vaccine safety is evaluated by identifying a target population, stratifying it based on receipt of the vaccine of interest, and comparing the incidence proportion and rate of specific AEFIs within each group. Crucially, adjusting for confounding factors is necessary to appropriately estimate a effect. In addition to confounding, we also need to be aware of selection bias due to discrepancies between target population and analysis population, loss-to-follow-up; and measurement error due to measurement of variables (Lash et al., 2021). In particular, we need to pay attention to the measurement bias of data from claims data (Yamana et al., 2017; Newcomer et al., 2018; Hempenius et al., 2021). This design enables the estimation of relative (ratio) effect measures and absolute (difference) effect measures. The assessment of rare AEFIs necessitates large cohorts.

SCCS design

SCCS design is a case-only design. The analysis population comprises individuals who have experienced an AEFI of interest. The observation period for individuals is segmented into a risk period and a control period, and the period during which the AEFIs occurred is utilized to estimate the incidence rate ratio. The risk period is defined as the period during which the AEFIs was considered to be clinically related to the vaccination. By creating within-individual comparisons, time-invariant confounders could be automatically adjusted for. Conversely, SCCS designs also have time-varying confounding (e.g., age in pediatric vaccines). Like cohort studies, SCCS studies should be alert to selection and measurement biases arising from the data. In particular, the SCCS target of estimation, the incidence rate ratio, is susceptible to measurement error related to the time of outcome occurrence and exposure initiation.

Power analysis in simulation

The power to detect rare AEFIs in a PASS was calculated for each design. For the cohort design, we calculated power considering different risk ratios, prevalences of the unvaccinated, matched ratio of vaccinated to unvaccinated, and varied sample sizes of vaccinated individuals. In this simulation, individuals who were not vaccinated were paired with vaccinated individuals randomly to define the observation period for the non-vaccinated, rather than to control for confounders. This ratio of pairs is called the “matched ratio” in this article. We estimated power by estimating p-values using the Fisher’s exact test. Note that to assess the behavior of power in scenarios devoid of bias in the estimates, we established our setting to be free from confounding. In parallel, for the SCCS design, we calculated power considering different rate ratio, observed period, risk period, and event counts. We derived p-values via Wald statistics and used the SCCS package in R (Weldeselassie, Whitaker & Farrington, 2022). For both designs, the null hypothesis was that relative ratio = 1, with two-sided significance level of 5%. The detailed settings for each design are summarized in Table 1, following the recommendations of Morris, White & Crowther (2019). Assuming that vaccination increases the risk of AEFIs, only the positive direction was evaluated. The number of simulation iterations was set to 5,000 and implemented using R version 4.3.0 (R Core Team, 2023). The code is archived at GitHub: https://github.com/ShuntaroS/23-power-cohort_SCCS-PASS.

Table 1 Setting of simulations for the cohort study design and the self-controlled case series design.

The simulation settings for each design are summarized as follows: aims, data-generating mechanisms, estimands, methods, and performance measures.

	Cohort study design	Self-controlled case series design	
Aims	Estimate the power under possible situations in post-authorization safety studies	
Data-generating mechanisms	•Effect measure (risk ratio): 1.2, 1.5, 2, 5	•Effect measure (rate ratio): 1.2, 1.5, 2, 5	
	•Prevalence of unvaccinated: 1/100000, 5/100000, 10/100000, 50/100000, 100/100000	•Observed period (years): 1, 2, 5, 10	
	•Sample size of vaccinated: 100000, 500000, 1000000, 2000000, 3000000, 4000000, 5000000	•Risk period (days): 14, 30, 42	
	•Matched ratio of vaccinated to unvaccinated: 1, 3, 5	•The number of events: 10 to 300 by 10	
Estimands	For example, risk ratios, risk differences, and odds ratios, but are not of interest in this simulation	For example, rate ratio, but is not be of interest in this simulation	
Methods	•Pvalue estimated by Fisher’s exact test	•Pvalue based on Wald statistics	
	•Null hypothesis: risk ratio = 1	•Null hypothesis: rate ratio = 1	
		•SCCS package of R was used	
Performance measure	Power with a threshold of 5% two-sided significance level	

Results

We estimated the power of the cohort study design and the SCCS design in likely scenarios for a PASS.

Cohort study design

Figure 2 illustrates the power trend. When the prevalence of the unvaccinated group was less than 1 in 100,000, even with a risk ratio of 2, a sample size of 100,000 vaccinated individuals did not demonstrate sufficient power to detect the effect. Assuming a prevalence of approximately 50/100,000, a near-null risk ratio may be detectable within a sample size of about 100,000. When the matched ratio of vaccinated to unvaccinated was altered to 1:1, 3:1, or 5:1, efficiency improved with a change in the ratio from 1:1 to 3:1. However, modifying the ratio from 3:1 to 5:1 did not increase efficiency.

Figure 2 Power of the cohort study design in the post-authorization safety studies setting.

Each data point represents the power in terms of interested AEFI prevalence in the unvaccinated group, risk ratio, sample size of the vaccinated group, and the ratio of the sample size in the unvaccinated group to the vaccinated group in the cohort study design. The line type, shape, or color of the line is the matched ratio of vaccinated to unvaccinated.

SCCS design

Figure 3 shows the trend of detection power for a risk period of 42 days. The power trend for a risk period of 14 days is shown in Fig. S1, and the power trend for a risk period of 30 days is shown in Fig. S2. The “saw-tooth phenomenon” was particularly pronounced when the observation period was longer than 3 years. The power did not increase with the expanded observation period, but rather decreased. Furthermore, when we observed the trend in power by risk period, power increased as the ratio of the risk period to the observation period became larger. When the incidence ratio was small, a substantial quantity of event cases was necessary.

Figure 3 Power of the self-controlled case-series design with 42-day risk period in the post-authorization safety studies setting.

Each data point represents the number of AEFI events of interest, the incidence ratio, and the power in terms of length of observation period in the self-controlled case-series design with 42-day risk period.

Discussion

In this study, we identified cohort study designs and SCCS study designs and their characteristics as common study designs used for PASS. In particular, we calculated the power of both cohort and SCCS designs under conditions in which a PASS is most applicable, especially in the context of rare AEFIs occurrences. In both designs, we found it necessary to estimate the sample size or number of cases to be used in the study based on a priori information and anticipated outcomes (Gault et al., 2017; Sultana et al., 2022).

Cohort study design

When the effect size is small and the non-vaccinated group’s prevalence is low, detecting any effect can be challenging with a realistically achievable sample size. In such cases, it is better to determine the sample size needed based on the clarity of the effect-size confidence interval before starting the study, rather than relying purely on hypothesis testing. If the calculated confidence interval falls within the pre-specified confidence interval, we may infer that there is likely no association between vaccination and the AEFIs under consideration (Lash et al., 2021). A Bayesian method could be considered if the statistical significance is not a concern in the frequentist statistical approach (Greenland et al., 2016).

In a cohort study design, evaluating the association between vaccination and an AEFIs involves several steps: handling data, matching of nonvaccinated individuals with vaccinated individuals for index date, and estimating with adjustments for confounders (using an outcome regression model or propensity score model (Hernán & Robins, 2020)). Since data handling and pairing demand significant computational resources, a ratio of three vaccinated individuals to each non-vaccinated individual is considered a good balance for efficiency.

Finally, we have consistently used the Fisher’s exact test for simulation of cohort studies, since the number of events may be extremely small in some cases. However, when the number of events is large, simulations based on the normal approximation should also be considered. The normal approximation method has better power than the Fisher’s exact test.

SCCS design

It was anticipated that a longer observation period would lead to a greater number of events, thereby increasing the power. However, findings revealed that the power actually increased with shorter observation periods. As such, the observation period should be set with consideration for the number of events that can be collected within that timeframe. A preliminary estimate of the number of events per observation period could assist in study design.

Focusing on the risk period, the power increased as the length of the risk period relative to the observation period increased. Since the length of the risk period is related to the AEFI setting, it is not desirable to easily vary the risk period. Thus, the ratio of the risk period to the observation period should also be taken into account in setting the observation period.

As the observation period lengthened, the instability of the power trend became more pronounced. This instability is referred to as the “saw-tooth phenomenon”. This phenomenon occurs in power calculations involving discrete data and was observed in SCCS analyses due to the discrete nature of the event counts both inside and outside the risk period (Farrington, Whitaker & Weldeselassie, 2018). Based on these results, we recommend that the power calculation not be performed only for a specific observation period and sample size; instead, both the observation period and sample size be varied over a somewhat range to check the transition of power.

One assumption of SCCS design is that within an individual, events can be recurrent and independent; non-recurrent events are rare (Farrington, Whitaker & Weldeselassie, 2018). As observation periods shorten, the chance of observing the AEFIs of interest only once in a PASS increases. In this case, the rarity of the event becomes important. Whitaker, Steer & Farrington (2018) reported that a cumulative incidence rate of less than 10% during the observation period would indicate a small bias, suggesting that in a PASS, the AEFIs of interest can be considered “rare”. However, the power may be unstable when the number of events is small. Thus, even when applying the SCSS, a cohort comprising 10 million individuals is required for an AEFIs with an incidence exceeding 100 events, which would correspond to the incidence of one event in every 100,000 individuals.

Comparison of two designs

In a PASS, we want to know not only the association between vaccination and AEFIs, but also the association of each vaccine if possible. Thus, confounding should be reduced as much as possible. Cohort study designs necessitate adjustment for numerous confounders. When events are infrequent, an outcome regression model might not adequately account for confounding. A propensity score model could sufficiently adjust for these factors, providing a valid effect estimate. However, because we are matching non-vaccinated persons against vaccinated persons, it may be preferable to take time into account instead of the usual propensity score estimation (Andersen et al., 2016). By contrast, an SCCS study can theoretically eliminate confounding, suggesting a high degree of validity in the effect estimation. Moreover, it is less computationally demanding than cohort studies, as the analytic population can be limited to instances where an outcome is present. Despite these advantages, a PASS based on this design requires a large cohort to ensure robust power due to the rarity of events.

Regarding estimable association and effect measures, cohort studies may be more beneficial for public health because SCCS studies can only estimate relative measures (incidence rate ratios). By contrast, cohort studies can estimate absolute measures (risk differences). Absolute and relative measures are both required when deciding whether to endorse the safety of certain vaccines.

In this study, we have evaluated the power of both designs based on hypothesis tests, but since the association between vaccination and AEFIs may well be close to null, a precision-based design or a Bayesian approach may also be helpful.

Strengths and limitations

This study’s strength is grounded in the evaluation of research design and power within a realistic PASS setting. However, it also presents certain limitations. First, we did not use a model assuming confounding structure in our simulations because we focused our attention only on the power transition. Second, in SCCS studies, we observed a decrease in power as the observation period increased. Further theoretical study of this behavior is needed.

Conclusions

In this study, we calculated the power of the cohort study and SCCS designs for scenarios in which a PASS might be employed, with a particular focus on the incidence of rare AEFIs. It may simply be necessary to use all the data available for either design. We found it necessary to estimate the sample size and number of events to be used in the study based on a priori information and expected results.

Supplemental Information

Figure S1 Power of the self-controlled case-series design with 14-day risk period in the Post-Authorization Safety Studies setting

Each data point represents the number of AEFI events of interest, the incidence ratio, and the power in terms of length of observation period in the self-controlled case-series design with 14-day risk period.

Click here for additional data file.

Figure S2 Power of the self-controlled case-series design with 30-day risk period in the Post-Authorization Safety Studies setting

Each data point represents the number of AEFI events of interest, the incidence ratio, and the power in terms of length of observation period in the self-controlled case-series design with 30-day risk period.

Click here for additional data file.

OpenAI’s ChatGPT was utilized for enhancing the language and expression of this manuscript.

Additional Information and Declarations

Competing Interests

Author Contributions

Data Availability

Shuntaro Sato reports personal fees from Nippon Boehringer Ingelheim Co., Ltd, outside the submitted work; Tomohiro Katsuta received payment for lectures from Merck Sharp & Dohme Corp. The other authers declare that they have no competing interests.

Shuntaro Sato conceived and designed the experiments, performed the experiments, analyzed the data, prepared figures and/or tables, authored or reviewed drafts of the article, and approved the final draft.

Yurika Kawazoe conceived and designed the experiments, authored or reviewed drafts of the article, and approved the final draft.

Tomohiro Katsuta conceived and designed the experiments, authored or reviewed drafts of the article, and approved the final draft.

Haruhisa Fukuda conceived and designed the experiments, authored or reviewed drafts of the article, and approved the final draft.

The following information was supplied regarding data availability:

The code is available at GitHub and Zenodo:

-https://github.com/ShuntaroS/23-power-cohort_SCCS-PASS.

-Sato S. 2023. Comparison design and Evaluation power in cohort and self-controlled case series designs for post-authorization vaccine safety studies. Zenodo. https://doi.org/10.5281/zenodo.10237514.

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
