# Peer review of "Comparison design and evaluation power in cohort and self-controlled case series designs for post-authorization vaccine safety studies"

_PeerJ, doi:10.7717/peerj.16780_

## Round 0.1 · original submission · Major Revisions

Thank you for choosing the PeerJ for this study. Your manuscript has been reviewed and assessed by two reviewers, and both of them agree with the fact that there are still a few points that need to be addressed. The comments of the reviewers are included at the bottom of this letter. Reviewers indicated that the introduction, methods, and results sections should be improved. We would be glad to consider a substantial revision of your work, where the reviewer’s comments will be carefully addressed one by one.

Reviewer 1 ·

Basic reporting

There was little background provided on power calculations for cohort and SCCS studies. Additionally, some of the citations could be updated. I have provided detailed feedback under “additional comments.”

Experimental design

The research question is a bit unclear. I recommend focusing this paper on the simulation and removing the comparison of study designs. The simulation set-up should be described in more detail and rational for the observation windows in the SCCS simulation should be provided.

Validity of the findings

The results should be discussed in the context of the current literature on sample size/power calculations. The authors provided their simulation code, which is a strength and allows for reproducibility.

Additional comments

This paper discusses cohort and self-controlled case series (SCCS) studies in the context of post-market vaccine safety studies. The authors conducted a simulation study to demonstrate power and sample size considerations between the two study designs.

An accessible discussion of power considerations in vaccine safety study with included code is a nice contribution to the literature. However, the pros and cons of cohort studies verses SCCS in vaccine research have been well enumerated in the literature. As such, I recommend focusing this paper on the sample size/power simulation. It would be helpful to move some of what is currently in the discussion on cohort and SCCS studies into the background section, and from there describe the simulations in the methods section. There is some causal language used in this paper that I recommend changing. Causality in vaccine safety studies requires strong assumptions and a discussion of the causal criteria. The authors should also consider removing the active and passive surveillance discussion because these study designs can be used in either type of data source.


Introduction

As mentioned above, I recommend reorganizing. I would remove the background on active and passive surveillance and instead introduce both study designs and provide background information on existing power/sample size considerations and work.

Lines 51-62 the authors state that passive surveillance cannot be used to assess a causal relationship, but that active surveillance with a cohort study or SCCS study can evaluate the causal relationship between vaccines and adverse events. I have concerns about this statement because it could be easily misinterpreted. Davis et al., 2005 does not discuss using active surveillance to assess causality; they discuss using active surveillance for better signal detection. I recommend removing all causal language throughout.

Line 72 Osokogu et al., 2016 is a bit dated and is a systematic review of safety literature in children, not specific to vaccines. Has the statistical power of PASS studies improved since 2016? Studies in children could face more challenges with sample size than more broad studies. I recommend identifying and citing a more recent systematic review of vaccine safety studies in a broader population.

Line 74-75 I do not think random sampling would ever be warranted in a vaccine safety study because the outcome is so rare. I would remove this statement because it is distracting from your message.

Materials and Methods

This section should be expanded. While the table describing the simulation set-up is helpful, the authors should consider including a text summary of the simulation set-up. Following a reorganized background, instead of summarizing cohort and SCCS study designs in this section, the authors could include a detailed description of the simulation set-up for each design.

In Table 1, for the SCCS design, how often are vaccine safety study observation periods 1 – 10 years? The results will be most useful with commonly used observation periods.

Lines 109-110 does confounding impact power in a cohort study?

Lines 86-87 case-control designs are less commonly employed in active surveillance as a reason for not including case-control in the simulations does not seem like good justification; I recommend removing the active surveillance component and providing a citation showing a decline in case-control designs for vaccine-safety studies if the field is moving away from them.

Results

This section should also be expanded. Figure 3 shows a saw-tooth phenomenon, which has been observed previously and should probably be discussed, please see: http://stats-www.open.ac.uk/sccs/thesis_patrick_musonda.pdf, section “4.4.3 Saw-tooth phenomenon”

Lines 124-127 Some of this will be more understandable for readers with a more detailed description of the simulation set-up in the methods section. Are the ratios for matched cohort designs?

Lines 133-134 “The power did not increase with the expanded observation period, but rather decreased.” Has this been shown in the literature before? I think this is part of the consideration when selecting appropriate risk periods.

Discussion

Lines 142-144 How did you determine it was necessary to estimate sample size a priori? Perhaps provide a citation that demonstrates the importance of determining sample size ahead of time.

Lines 163-164 “It was anticipated that a longer observation period would lead to a greater number of events, thereby increasing power. However, findings revealed that the power actually increased with shorter observation periods.” Related to an above comment – I think this is part of why careful selection of time periods is important for this design.

Lines 174-175 “While bias does not pose an issue in this context” what bias and what context? It would be helpful to be explicit.

Line 196-197 please provide a citation

Lines 204-207 You could add a discussion of selection bias and measurement error in terms of the two study designs; it does not have to be a limitation. Additionally, I don’t think it was clear in the comparison of study designs that both have issues with confounding. For the SCCS, the only confounding is time-varying. There was no confounding structure to the simulations, and perhaps that is a limitation.

Reviewer 2 ·

Basic reporting

I believe there are a few typos in the manuscript.

1. Last sentence in Figure 2 caption: "the ratio of" was repeated.

2. In Figure 2, the graph is colored by "ratio of vaccinated to unvaccinated". In Table 1 it also says "ratio of vaccinated to unvaccinated". However it conflits with the last sentence in Figure2 caption and line 125 on page 8 where both described it as "ratio of unvaccinated to vaccinated". Please keep the definition consitent throughout.

Experimental design

No comment

Validity of the findings

1. The number of simulation runs is 1000. I would suggest using a larger number of iterations, e.g., 5000 runs in order to get a better estimation of the statistical power.

2. Fisher's exact test was used to compare proportions in two samples. The authors didn't make the hypothesis testing clear enough. Is the proportion ratio (p1/p2, pi being the the incidence rate in the ith group) compared to 1? What was the margin used in the test? When we are comparing the ratio of two extremely small proportions, a clinically meaningful effect size (ratio) should be defined and used as the margin in the test. Otherwise with such a big sample size we might end up declaring two very close proportions as statistically different. It might also be intereting to evaluate the impact of the margin on the statistical power.

3. The sample size in these studies are fairly large, why not use normal approximation instead of fisher's exact test?

Additional comments

no comment

---

## Round 0.2 · accepted · Accept

The authors addressed the reviewers' concerns and substantially improved the content of the manuscript. So, based on my assessment as an academic editor, the manuscript can be accepted in its current form.

Reviewer 2 ·

Basic reporting

The authors addressed all my questions. The manuscript reads well now.

Experimental design

no comment

Validity of the findings

no comment

Additional comments

no comment